



# Development of CAS-ESM_MMF: Improving East Asian Summer Precipitation Simulation with a Multiscale Modeling Framework

†Guangxing Lin[1,2], Wei Liao[2,3], Zhaohui Lin[2,3], He Zhang[2,3], Wenbin Kou[4], Xiaojie Guo[2,3], Zhenghui Xie[2,3], Qiu Yang[5], Chenglai Wu[2,3], and Minghua Zhang[6]

[1]College of Ocean and Earth Sciences, Xiamen University, Xiamen, 361005, China.
[2]State Key Laboratory of Earth System Numerical Modeling and Application, Institute of Atmospheric Physics, Chinese Academy of Sciences, Beijing, 100029, China.
[3]College of Earth and Planetary Sciences, University of Chinese Academy of Sciences, Beijing, 100049, China
[4]Frontiers Science Center for Deep Ocean Multispheres and Earth System (FDOMES) and Key Laboratory of Marine
Environmental Science and Ecology (Ministry of Education), Ocean University of China, Qingdao, 266100, China.
[5]Department of Atmospheric and Oceanic Sciences, School of Physics, Peking University, Beijing, 100871, China
[6]School of Marine and Atmospheric Sciences, Stony Brook University, Stony Brook, NY, 11794-5000, USA

*Correspondence to*: Guangxing Lin (linguangxing@xmu.edu.cn)

**Abstract.** Traditional global climate models (GCMs) exhibit substantial biases in simulating precipitation over East Asia,
largely due to uncertainties in convection parameterizations. To address this issue, we implement a Multiscale Modeling Framework (MMF), which explicitly resolves convection in a cloud resolving model, into the atmospheric component of the Chinese Academy of Sciences Earth System Model (CAS-ESM). Simulations using CAS-ESM with and without MMF reveal that the MMF implementation significantly reduces the wet bias around the Tibetan Plateau and the dry bias over South China and Southeast Asia. The intensity–frequency characteristics of precipitation are more realistically represented in
the MMF version. In addition, the CAS-ESM with MMF better captures the monthly evolution of precipitation and simulates a more realistic seasonal migration of the East Asian rainband, albeit with a somewhat step-wise progression. Further enhancement is achieved by incorporating a convective momentum transport (CMT) parameterization, typically neglected in previous MMF implementations. This inclusion leads to a smoother northward migration of the rainband, more consistent with observations. Comparison with ERA5 reanalysis suggests that this improvement is associated with a more accurate
simulation of the western Pacific subtropical high. These results demonstrate that MMF, especially when combined with CMT, substantially improves the simulation of East Asian precipitation. This modeling advancement offers a promising approach for evaluating regional precipitation responses to future climate change.

## 1 Introduction

Traditional global climate models (GCMs), such as those participating in the Coupled Model Intercomparison Project
(CMIP), exhibit substantial biases in simulating precipitation over East Asia. For example, CMIP5 models consistently overestimate mean precipitation over South China and the eastern Tibetan Plateau (Chen and Frauenfeld, 2014; Kusunoki and Arakawa, 2015; Xin et al., 2020; Xu et al., 2022; Zhang and Chen, 2016). Although these biases are partially alleviated in CMIP6, they remain prominent (Cui et al., 2021; Peng et al., 2025; Xin et al., 2020; Xu et al., 2022). In addition, most



models overestimate the frequency of light rainfall and underestimate the intensity of heavy rainfall events (Sun et al., 2006;

Stephens et al., 2010). They also show large biases in simulating extreme precipitation events across East Asia (Kim et al., 2019) and exhibit considerable spread among model members. Moreover, many models perform poorly in capturing the seasonal evolution of the East Asian monsoon, particularly the northward migration of the summer rainband (Kusunoki and Arakawa, 2015). These deficiencies in traditional GCM models pose severe challenges to accurately predicting precipitation and its seasonal variations in East Asia.

Many of these biases arise from the coarse grid spacing (around 100 km) of traditional GCMs and the inherent limitations of subgrid-scale cloud and convection parameterizations that have significant uncertainties. First, convection triggering is often based on empirical formulations, which may inadequately capture the complex physical processes governing convective initiation (Randall et al., 2016; Zhang et al., 2021; Villalba-Pradas and Tapiador, 2022). Furthermore, the weak coupling between convection and the planetary boundary layer in many models limits the feedback between boundary-layer

thermodynamics and cloud development, contributing to biases in low cloud and vertical moisture transport (Brient et al., 2019). Most importantly, conventional convective parameterizations often assume convective quasi-equilibrium, the notion that convective processes quickly adjust to large-scale forcing and maintain a balance state (Craig and Cohen, 2006; Yano and Plant, 2012). However, recent studies suggest that convective quasi-equilibrium holds only during intense, deep convection events and may not be valid under broader atmospheric conditions (Li et al., 2024). These shortcomings highlight

the need for more physically based representations of convection.

Different from the traditional GCMs, the multiscale modeling framework (MMF), also known as super-parameterization (SP), uses a cloud resolving model to explicitly simulate the convection, bypassing the convection parameterization (Grabowski, 2001). MMF embeds a cloud-resolving model (CRM) within each grid cell of a host-GCM (Khairoutdinov et al., 2005; Randall et al., 2003), typically using ~1 km horizontal resolution and two-dimensional grid structure (one horizontal

and one vertical dimension). This hybrid approach is two orders of magnitude less computationally expensive than full global convection-permitting models (Grabowski, 2001). MMF has been shown to improve the spatial distribution of climatological precipitation (Khairoutdinov et al., 2005, 2008; Hannah et al., 2020), tropical intraseasonal variability including the Madden–Julian Oscillation (Benedict and Randall, 2009), and the diurnal cycles of precipitation (Pritchard and Somerville, 2009). It also improves the representation of mesoscale convective systems and associated rainfall (Lin et al.,

60 2019, 2022).

Despite these advancements, a key physical process, convective momentum transport (CMT), has often been neglected in MMF applications. CMT refers to the interactions between clouds and large-scale circulation in the form of vertical transport of horizontal momentum by convective updrafts and downdrafts (Moncrieff, 1992; Moncrieff et al., 2017). Because the embedded CRMs in MMF are typically two-dimensional, the cloud within CRMs can directly interact only with large-scale

wind component parallel to the CRM columns, leaving the perpendicular component unresolved (Tulich, 2015). Neglecting the unresolved CMT component could substantially degrade model performances (Cheng and Xu, 2014). To address this limitation, Tulich (2015) developed a parameterization scheme, called the Explicit Scalar Momentum Transport (ESMT)





scheme, for representing CMT in 2D CRMs. Yang et al. (2022) validated this approach, showing that it reproduces CMT fields comparable to those generated by fully three-dimensional CRMs. Yet, the influence of CMT on large-scale circulation
and seasonal precipitation patterns over East Asia remains unexplored.

The Chinese Academy of Sciences Earth System Model (CAS-ESM), developed by Institute of Atmospheric Physics (IAP), is a CMIP6-class model that also exhibits many of the aforementioned shortcomings in simulating East Asian precipitation. For instance, Su et al. (2014) found that the atmospheric model of CAS-ESM1.0 (IAP-AGCM4.0) simulates a weaker-than-observed East Asian summer rainband and fails to reproduce its northward migration. For the atmospheric model of CAS-
ESM2.0 (IAP-AGCM5.0), there also exists significant dry bias for summer monsoon rainfall in South China (LIN Zhaohui et al., 2025; Zhang et al., 2022). Although Zhang and Chen (2016) demonstrated that the employment of MMF to CAM5.2 improves East Asian precipitation simulation, their study did not evaluate the role of CMT and was conducted at a coarser grid spacing of ~2°. Therefore, this study aims to implement MMF in the atmospheric component of CAS-ESM (IAP-AGCM4.1) with a higher horizontal resolution of ~1° to improve East Asian precipitation simulation and evaluate the impact
of including CMT scheme on the modeling northward movement of summer East Asian precipitation.

The objectives of this study are threefold: (1) To document the implementation of MMF within the CAS-ESM framework; (2) To evaluate its impact on the simulation of East Asian precipitation; and (3) To investigate the role of CMT parameterization in improving the simulation of the seasonal northward migration of the East Asian summer rainband. The remainder of this paper is organized as follows. Section 2 describes the model, datasets, experimental design, and analytic areas. Section 3
evaluates the improvements in precipitation simulation from MMF relative to traditional convection schemes and assesses the role of CMT. Section 4 summarizes and discusses the main findings.

## 2 Methods

### 2.1 Development and description of CAS-ESM

We develop an MMF version of CAS-ESM, referred to CAS-ESM_MMF for short hereafter, by coupling the IAP-AGCM
and a CRM. We follow the approach by Grabowski (2004) that was also used in Benedict and Randall (2009), Khairoutdinov and Randall (2001), and Khairoutdinov et al. (2005). The general idea is that a two-dimensional CRM is embedded within each grid column of the IAP-AGCM, enabling the explicit representation of subgrid-scale cloud and convective processes. For each model time step, a large-scale forcing from IAP-AGCM is introduced to the CRM, which keeps the states of the two models consistent with each other, while that the effects of clouds, turbulence, and convection,
which are simulated in the CRM, are fed back to the global model.

The dynamical core of IAP-AGCM4.1 is based on a finite difference method that uses transformed velocity as prognostic variables (IAP transform) (Zhang et al., 2020). The model uses standard latitude-longitude grid with a horizontal resolution of 1° (approximately 100 km). Vertically, the model has 30 layers, with the model top situated at 2.2 hPa. The GCM physics time step is 30 minutes. Compared to its predecessors (IAP-AGCM4.0), IAP-AGCM4.1 incorporates the physical





parameterization package of the Community Atmospheric Model version 5 with some modifications to suit its dynamical

core (Adeniyi et al., 2019; Zhang et al., 2020). The shallow convection scheme is UW diagnostic Turbulent Kinetic Energy

(TKE) scheme (Bretherton and Park, 2009; Park and Bretherton, 2009). The parameterization for deep convection originates

from Zhang and McFarlane (1995), incorporating subsequent modifications as detailed by Neale et al. (2008) and Richter

and Rasch (2008). For cloud processes, the macrophysics scheme follows Park et al. (2014), while the microphysics are

represented by the two-moment scheme of Morrison and Gettelman (Morrison and Gettelman, 2008).

The embedded CRM is the System for Atmospheric Modeling, developed by Marat Khairoutdinov and David Randall

(Khairoutdinov and Randall, 2003). This CRM employs the anelastic approximation equations. To reduce computational

expense, the CRMs adopted 2D grid structure, oriented in the zonal (east-west) and vertical directions. The CRM comprises

32 horizontal columns horizontally with a grid spacing of 4 km. Vertically, the CRM shares the lowest 28 levels with the

host-GCM. The CRM integration time step is 20 seconds. Given the CRM's 4-km grid spacing, which permits the explicit

representation of deep convection, cumulus parameterization is no longer needed within the CRM. Instead, subgrid-scale

processes are represented as follows: cloud microphysics is handled by the two-moment scheme from Morrison et al. (2005),

and turbulence is parameterized using a 1.5-order closure scheme based on prognostic turbulent kinetic energy

(Khairoutdinov and Kogan, 2000). Radiative transfer is calculated using the Rapid Radiative Transfer Model for GCMs

(Mlawer et al., 1997; Iacono et al., 2008), and aerosols are represented via the Modal Aerosol Model with 3 modes(Liu et al.,

2012). To ensure conservation of mass and energy, periodic lateral boundary conditions are applied to each CRM domain in

the horizontal direction.

A notable limitation of the 2D CRM configuration is its inherent inability to directly represent momentum feedback , namely

the convective momentum transport (CMT) (Cheng and Xu, 2014; Khairoutdinov et al., 2008; Tulich, 2015), thus the typical

MMF neglects the CMT. To account for the impact of CMT on the simulation of East Asian precipitation, a subsequent

experiment incorporates a CMT parameterization, using the ESMT scheme developed by Tulich (2015), in the IAP-MMF

(CAS-ESM_MMF). The adopted ESMT scheme draws on the approach of CMT parameterization in traditional GCMs. It

diagnoses the pressure gradient force perpendicular to the CRM columns by solving a linearized Poisson equation, while the

pressure gradient force within the CRM columns is explicitly calculated by the model.

**2.2 Experimental design**

Three numerical experiments are conducted using prescribed sea surface temperatures (SSTs). The SST dataset represents

the 1995–2005 climatology provided by the Hadley Centre (Rayner et al., 2003). The simulation framework with fixed SST

was chosen partly to reduce the need for longer simulations to account for inter-annual variability. The simulations are

initialized on January 1, 2000, and integrated for 7 years. The first year of integration is discarded as model spin-up, with the

subsequent 6 years used for analysis. The experiments are set as follows (Table 1):



Table 1: Experiments design

| Experiments | Deep Convection | Convective Momentum Transport |
| --- | --- | --- |
| CAS-ESM | ZM scheme | YES (Richter and Rasch, 2008) |
| CAS-ESM_MMF | CRM resolved | NO |
| CAS-ESM_MMF_MF | CRM resolved | YES (Tulich, 2015) |

Since the MMF simulation is much more expensive than CAS-ESM, the third set of experiment only run 4 years initiating on January 1, 2000 to demonstrate CMT effects on East Asia summer precipitation simulations.

**2.3 Observation and reanalysis datasets**

Model results are evaluated against several observational and reanalysis datasets. Monthly precipitation data from the Global Precipitation Measurement (GPM) mission (IMERG Final Run V06B L3) 0.1° monthly precipitation rate (Huffman et al., 2020) and the Global Precipitation Climatology Project (GPCP V2.3) 2.5° monthly precipitation rate (Adler et al., 2018) are used to assess the simulated climatological mean state and annual cycle characteristics of East Asian precipitation. Daily

precipitation data, sourced from the GPM (0.1°) and the GPCP 1-degree daily (1dd v1.3) product (Huffman et al., 2001), are employed for analyzing seasonal variations. Hourly precipitation data, derived from GPM (0.1°) half-hourly rain rate, are utilized to compute probability distribution functions (PDFs). Other meteorological variables for evaluating large-scale circulation features are obtained from the 0.25° fifth-generation ECMWF atmospheric reanalysis (ERA5) (Hersbach et al., 2020). All datasets cover the same time period as the model simulations.

**2.4 Analytic areas**

This study focuses on the simulation of precipitation over East Asia, especially eastern China. Given the significant regional differences in precipitation characteristics across eastern China, five representative subregions are selected to investigate the temporal variability of precipitation as well as its frequency-intensity relationship in greater detail: South China (23° N–26° N, 110° E–116° E), Eastern Tibetan Plateau (27° N–32° N, 100° E–106° E), Northeast China (40° N–50° N, 120° E–130° E),

the middle and lower reaches of the Yangtze River Basin (28° N–32° N, 113° E–120° E), and North China (34° N–40° N, 113° E–117° E) (Fig. 1). These five selected subregions span from low to mid–high latitudes and from low-altitude to high-altitude areas, representing the typical climatic regimes of eastern China.



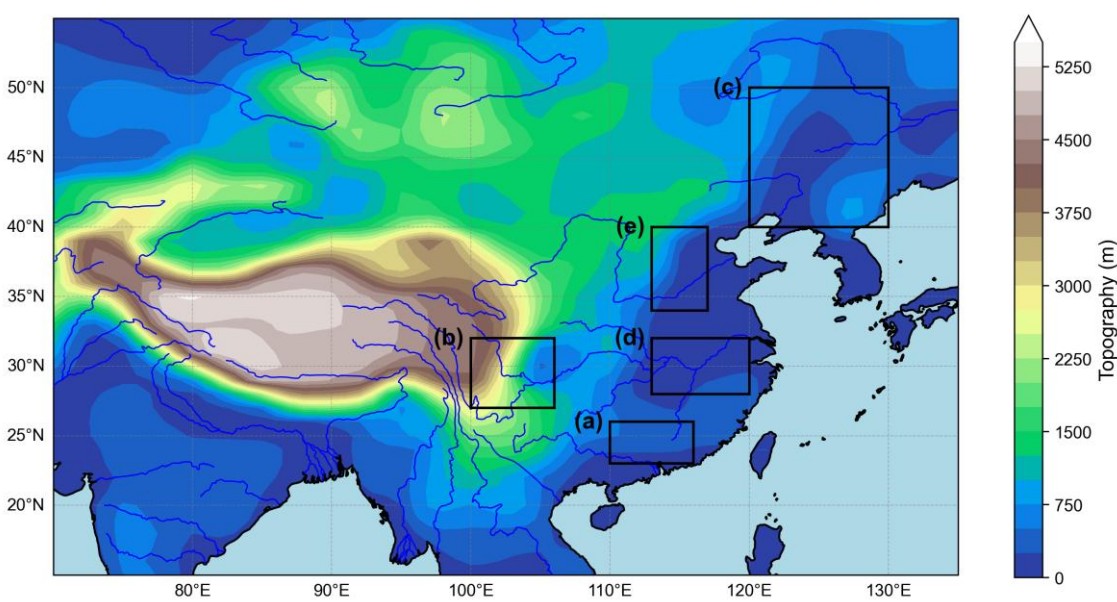

**Figure 1. Model topography and the geographical locations of five subregions: (a) South China (23° N–26° N, 110° E–116° E), (b) Eastern Tibetan Plateau (27° N–32° N, 100° E–106° E), (c) Northeast China (40° N–50° N, 120° E–130° E), (d) the middle and lower reaches of the Yangtze River Basin (28° N–32° N, 113° E–120° E), and (e) North China (34° N–40° N, 113° E–117° E)**

## 3 Results

### 3.1 Climatology of global precipitation

We begin by examining the climatological mean precipitation during the 2001–2006 simulation period. Figure 2 compares the observed annual mean precipitation (GPM, Fig. 2a) with simulations from CAS-ESM (Fig. 2b) and CAS-ESM_MMF (Fig. 2c), limited to 60° S–60° N due to GPM data sparsity at high latitudes. Both models reproduce the general spatial distribution, though they underestimate global mean precipitation relative to observations.

CAS-ESM shows prominent dry biases over South Asia, the Western Pacific, and northern South America (Fig. 2b), where precipitation is largely associated with mesoscale convective systems (Feng et al., 2021). These biases are substantially reduced in CAS-ESM_MMF (Fig. 2c), suggesting improved representation of convective processes, consistent with prior MMF studies (Lin et al., 2019, 2022). Additionally, the classic "double Intertropical Convergence Zone" bias present in CAS-ESM is notably mitigated in CAS-ESM_MMF (Fig. 2b vs. Fig. 2c), similar improvement has been found in other super-parameterized (SP) models such as SP-CAM (atmospheric component of CESM-MMF; Khairoutdinov et al., 2005), SP-E3SM (Hannah et al., 2020), and SP-MIROC (Yamazaki and Miura, 2024). However, CAS-ESM_MMF introduces wet biases over Central Africa and dry biases over the tropical Pacific and mid-to-high-latitude oceans. Overall, the MMF improves the simulation of convective precipitation on a global scale, though some systematic errors remain.

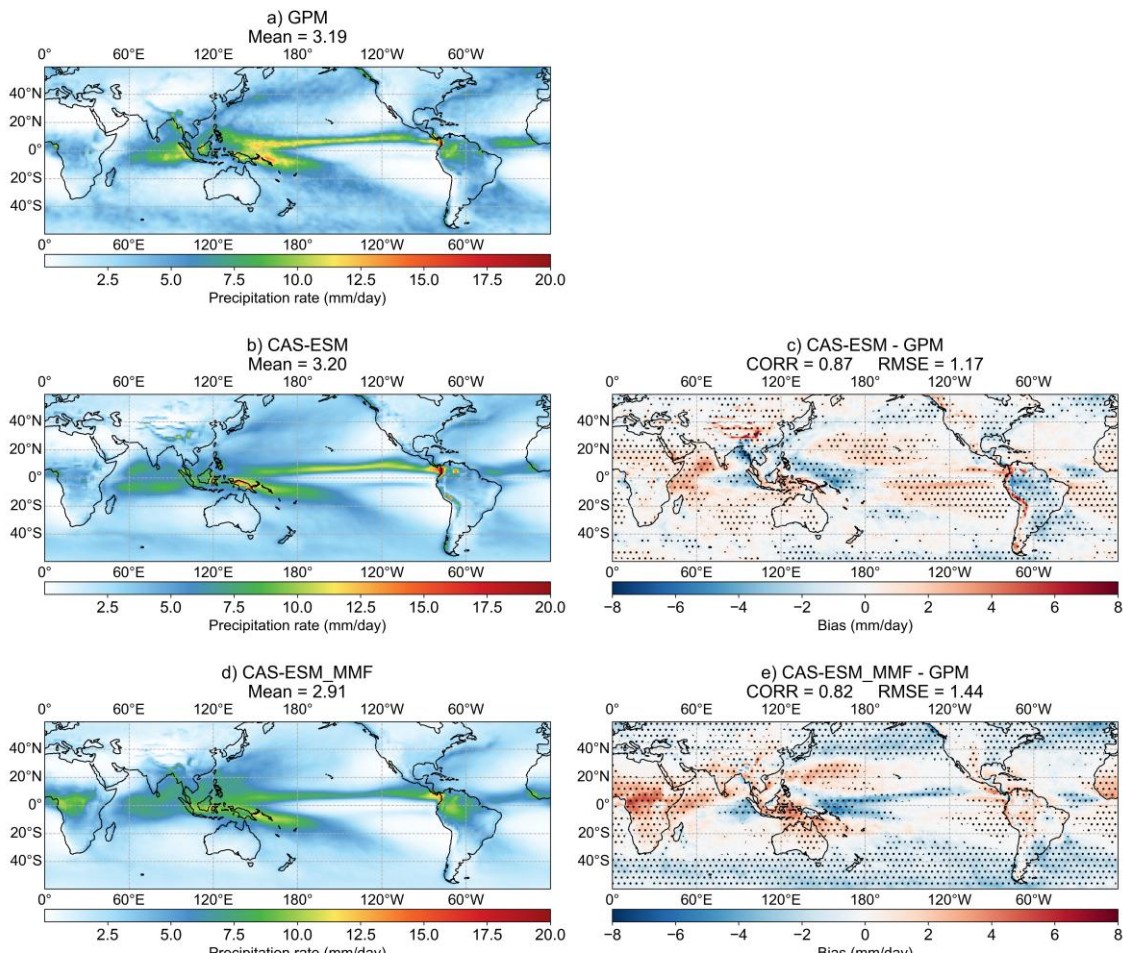

**Figure 2. Annual mean precipitation (mm/day) for 2001-2006 from (a) GPM observation, (b) CAS-ESM, and (d) CAS-ESM_MMF models. Panels (c) and (e) show the model biases relative to GPM. Area-weighted global means are noted in (a, b, d). Spatial correlation (CORR) and root mean square error (RMSE) are shown for bias plots. Stippling indicates where the bias is statistically significant (p < 0.05, Student's t-test).**

## 3.2 Summer precipitation over East Asia

This section focuses on the summer (June–July–August, JJA) precipitation over East Asia, with particular emphasis on the China region. Figure 3 presents GPM observations, model-simulated JJA mean precipitation, and the corresponding biases.



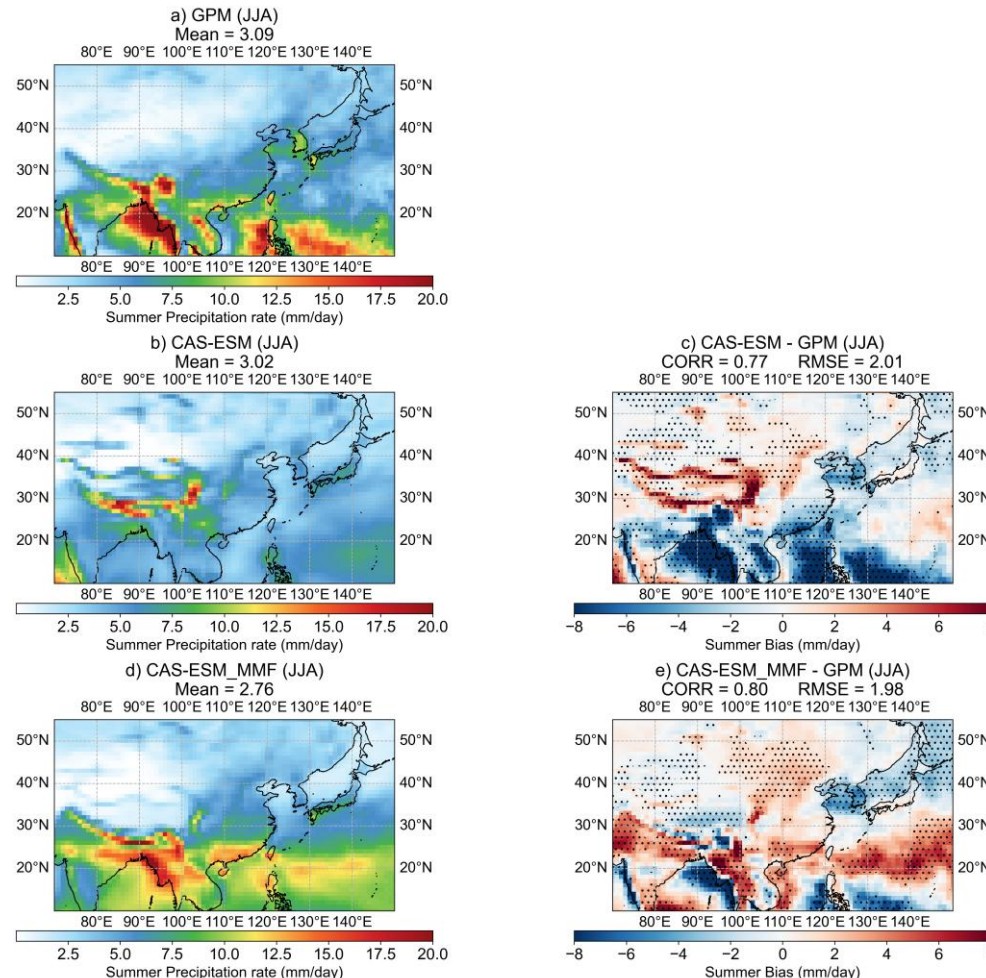

**Figure 3. Similar to Figure 2 but averaged over boreal summer months (June, July, and August) and focus on East Asia. (unit: mm/day)**

As shown in Figs. 3a and 3b, the CAS-ESM simulation exhibits substantial biases in both spatial distribution and regional mean precipitation compared to observations. The model overestimates precipitation over the Tibetan Plateau while underestimating it over the Bay of Bengal and the Western Pacific (Fig. 3c). These deficiencies highlight its limited ability to capture the key spatial features of East Asian summer rainfall. In contrast, the CAS-ESM_MMF, which replaces the conventional convection parameterization with an embedded two-dimensional cloud-resolving model (2D CRM) in each grid cell, shows substantial improvements in both pattern and magnitude (Fig. 3d). The CAS-ESM_MMF better reproduces the observed precipitation distribution, although some biases persist in the intensity and location of precipitation centers (Fig. 3e). Notably, the dry bias over the Bay of Bengal is significantly reduced, while slightly wet biases appear over the Indian Peninsula, Indochina Peninsula, South China, and the Western Pacific. Quantitatively, the CAS-ESM_MMF achieves higher spatial correlation with observations and a lower root mean square error (RMSE) compared to CAS-ESM. Similar



improvements are evident in the annual mean precipitation over East Asia (figure not shown). Overall, the CAS-ESM_MMF substantially improves the simulation of East Asian summer precipitation, primarily due to its enhanced representation of convective processes.

### 3.3 Annual cycle and PDF of regional precipitation

Figure 4 shows the annual cycle of regionally averaged precipitation for these subregions. Overall, CAS-ESM_MMF (red line) more closely matches observations (black and brown lines) than CAS-ESM (blue line) across all five regions. In South China, CAS-ESM significantly underestimates precipitation, particularly during summer, and fails to reproduce the observed seasonal progression. While the observation shows a single peak in June, CAS-ESM simulates two peaks in May and July. In contrast, CAS-ESM_MMF better captures the single-peak feature, though it overestimates July and August precipitation and delays the peak by about one month. Over the eastern Tibetan Plateau, CAS-ESM consistently overestimates precipitation throughout the year, especially in summer. CAS-ESM_MMF reduces this overestimation and better reproduces the observed seasonal trend. In Northeast China, CAS-ESM overestimates precipitation during spring and autumn, whereas CAS-ESM_MMF reduces these biases, resulting in a closer match to observations. For the middle and lower reaches of the Yangtze River and North China, both models show biases relative to observations, but CAS-ESM_MMF generally performs slightly better than CAS-ESM.

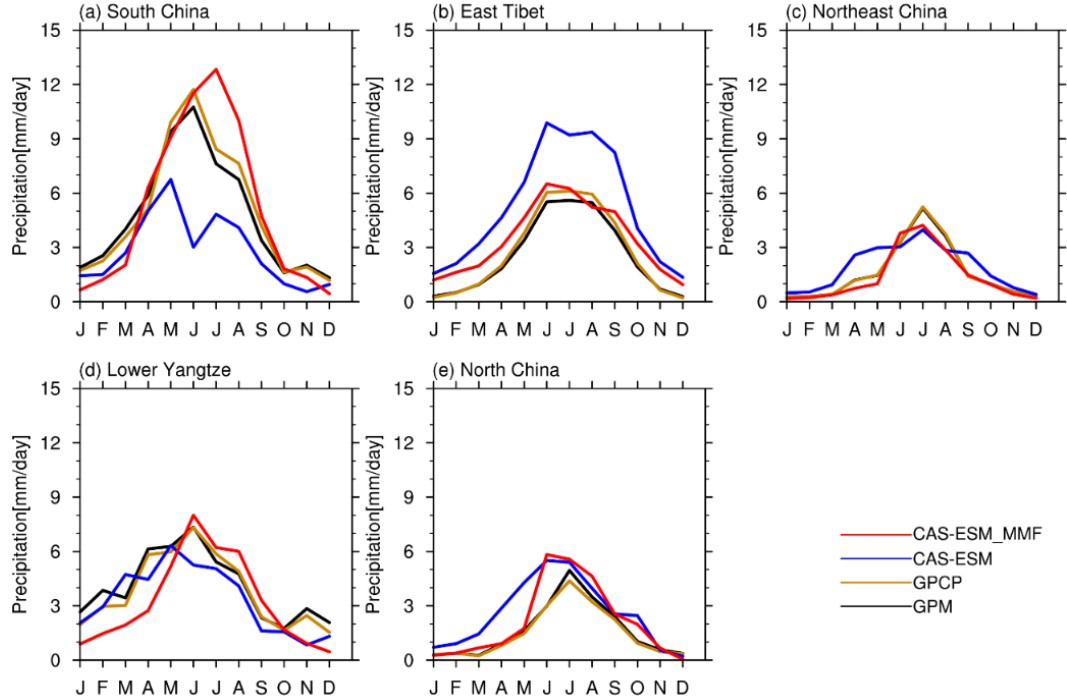

**Figure 4. Annual cycle of precipitation rate (units: mm/day) averaged over the five subregions shown in Figure 1.**




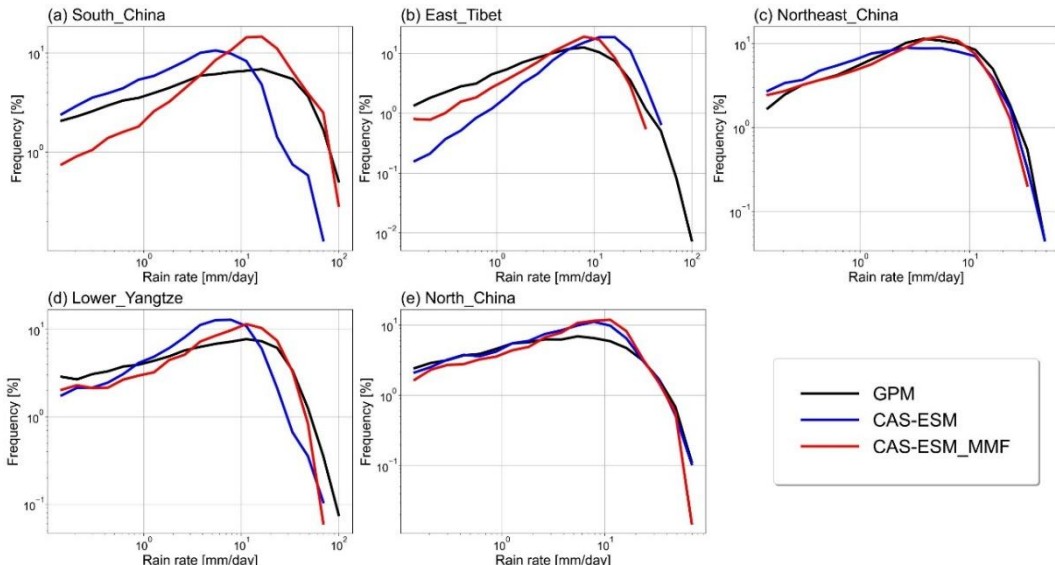

210

**Figure 5. Probability distribution function (PDF) of observed and simulated summer hourly precipitation rate (unit: mm/day) averaged over the five subregions. The locations are annotated on the top left of each subplot.**

To further explore the sources of mean precipitation biases, we examine the probability density functions (PDFs) of hourly precipitation for these five subregions (Fig. 5). Note that the original GPM data have a temporal resolution of half an hour. In this study, two consecutive half-hourly precipitation values are averaged to produce hourly data. In South China and the middle and lower reaches of the Yangtze River Basin, regions strongly influenced by the East Asian monsoon, CAS-ESM exhibits a typical bias seen in many traditional climate models (Sun et al., 2006; Zhang and Chen, 2016), with excessive light rain and an underrepresentation of heavy rainfall. The complex interactions between the moist monsoon air masses and continental air masses in these regions favor mesoscale convective systems, which contributes significantly to the total precipitation in these regions during summer (Feng et al., 2021). The inability of traditional convection parameterization schemes to adequately simulate such processes explains the underestimation of summer mean precipitation in CAS-ESM (Fig. 3c). By explicitly resolving convections, CAS-ESM_MMF substantially improves the heavy precipitation simulation, leading to a better agreement with observed JJA mean precipitation (Fig. 3d). However, CAS-ESM_MMF tends to underestimate light rain frequency but overestimate moderate-intensity rainfall (Figs. 5a and 5d). While the light rain bias has limited impact on the total precipitation, the excess moderate-intensity rainfall likely contributes to the wet bias in JJA mean precipitation (as seen in Figs. 4a and 4d). In the Eastern Tibetan Plateau, CAS-ESM overestimates moderate-to-heavy rainfall, resulting in an overestimation of JJA mean precipitation. CAS-ESM_MMF reduces the frequency of heavy rainfall events, improving the JJA mean, but the overall PDF still deviates substantially from GPM observations. In Northeast China and North China, both models' precipitation PDFs align reasonably well with observations, and their simulated JJA mean precipitation is also relatively accurate (Figs. 4c and 4e).



## 3.4 Seasonal movement of the rainband over eastern China

Figure 6 displays the monthly mean precipitation from 2001 to 2006, comparing observations and model simulations to examine the intraseasonal evolution of precipitation over East Asia. To ensure a fair comparison, the GPM precipitation data are interpolated onto the model grid using conservative interpolation, while the GPCP data (1°×1° resolution) are directly

235 compared to the model output. The two observational datasets show consistent spatial patterns of precipitation, though they differ in intensity; these inter-observational differences are much smaller than the discrepancies between models and observations. Both observations show (Figs. 6a and 6b) that precipitation concentrates in three main regions: eastern China, the eastern coast of the Bay of Bengal, and the southern foothills of the Himalayas. The latter two regions are primarily influenced by the South Asian summer monsoon, with precipitation increasing in May and decreasing in September. CAS-

240 ESM (Fig. 6c) severely underestimates precipitation over these two regions, consistent with the common dry bias in traditional parameterized GCMs (Goswami and Goswami, 2017). This dry bias is substantially alleviated in CAS-ESM_MMF (Fig. 6d).

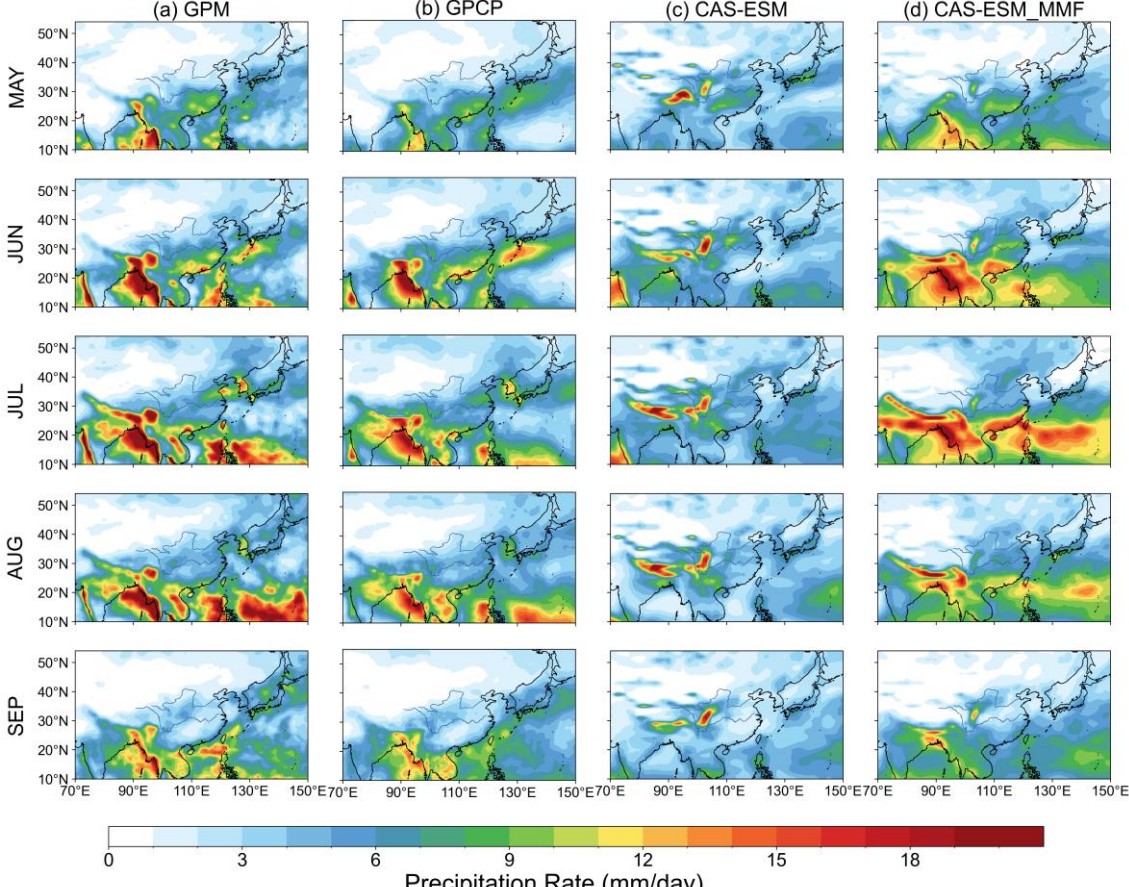





**Figure 6. Spatial distribution averaged from 2001 to 2006 of each wet season months rainfall (unit: mm/day) over East Asia. From the first row to the last row correspond to May to September. The first and second columns represent (a) GPM and (b) GPCP observations respectively, and the last two columns show the results from (c) CAS-ESM and (d) CAS-ESM_MMF.**

Over eastern China, the East Asian Summer Monsoon drives a southwest- northeast- oriented rainband that migrates northward with the seasonal progression of the Western Pacific Subtropical High (Ding, 1992; Sampe and Xie, 2010). Observations (Figs. 6a and 6b) show the rainband located over South China in June, moving to the Shandong Peninsula and the Korean Peninsula in July. CAS-ESM (Fig. 6c) fails to simulate this rainband and its seasonal movement, with consistently underestimated precipitation over southern China. In contrast, CAS-ESM_MMF (Fig. 6d) reproduces the northward migration, though the rainband appears over the Yangtze River Basin in June and reaches North China by July, approximately one month earlier than observed. Additionally, the rainband advances too rapidly, overshooting its observed progression. Despite these biases, the simulation of the rainband and its seasonal migration is substantially improved in CAS-ESM_MMF relative to CAS-ESM. However, CAS-ESM_MMF may produce overly strong convection in some regions, such as South China in July and the Indochinese Peninsula in June-July.

The northward migration of the rainband is further illustrated by latitude–time Hovmöller diagrams of pentad-averaged (5-day mean) zonal mean precipitation over eastern China (20°–45° N, 110°–130° E) in Figure 7. Observations (Figs. 7a, 7b) show precipitation concentrated south of 30° N before the 31st pentad (early June), with a gradual northward shift reaching 35° N (Huang–Huai River Basin, region between the Yellow River and the Huai River in eastern China) in July and extending to around 40° N by August. CAS-ESM (Fig. 7c) fails to reproduce this northward migration, with both precipitation intensity and movement poorly represented. In contrast, CAS-ESM_MMF (Fig. 7d) captures the general northward advance but with a faster progression and smaller rainfall magnitude in North China than observed. Specifically, the rainband's northward migration in CAS-ESM_MMF occurs 3–4 pentads (15–20 days) earlier than in observations, and the transition is abrupt, with the rainband leaping from ~34° N to ~42° N around the 33rd pentad. Overall, CAS-ESM_MMF significantly improves the simulation of the rainband's seasonal migration compared to CAS-ESM, though timing and migration speed biases remain.

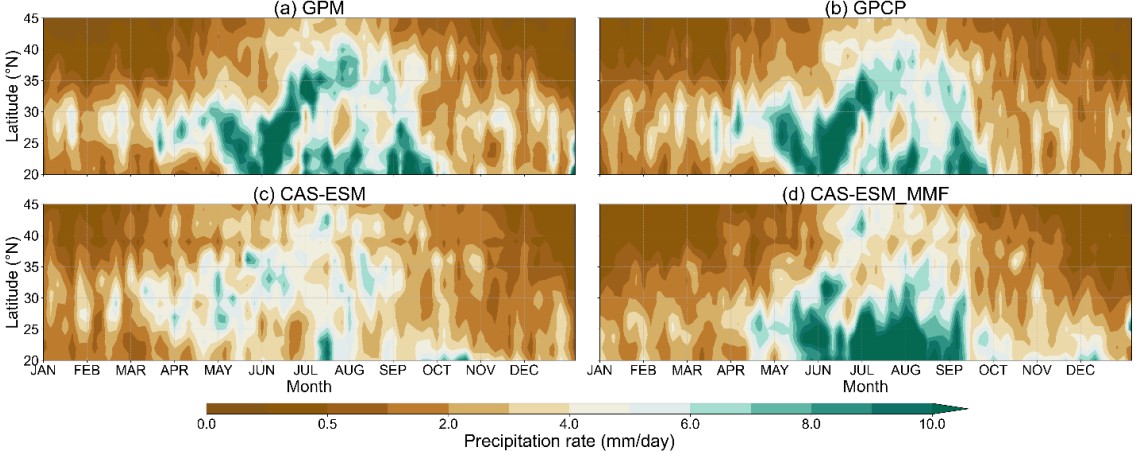



**Figure 7. Latitude-time Hovmöller diagrams of pentad (5-day) mean precipitation (unit: mm/day) averaged along the 110°-130° E over eastern China (20°-45° N, 110°-130° E) from (a) GPM, (b) GPCP-1dd, (c) CAS-ESM, (d) CAS-ESM_MMF. Here we calculate daily climatology from 2001 to 2006, then smoothen daily data to pentad data.**

Given the critical role of the Western Pacific Subtropical High (WPSH) in influencing the rainband's movement (Ding, 1992; Sampe and Xie, 2010), we further examine the large-scale circulation patterns. Figure 8 shows the 500 hPa wind and geopotential height fields over East Asia during JJA. Observations indicate that the WPSH strengthens and extends westward in June, then weakens and retreats eastward in August, with its ridge line gradually migrating northward, modulating the rainband's northward shift (as seen in Fig. 6). CAS-ESM simulates an excessively strong WPSH, with a westward-biased boundary dominating the southeastern coast of China, contributing to the persistent dry bias over southern China during summer. CAS-ESM_MMF, by contrast, simulates a weaker WPSH, alleviating the dry bias over southern China. However, the WPSH weakening is overestimated in CAS-ESM_MMF, resulting in an eastward-biased ridge position relative to observations in July. Compared to observation, CAS-ESM_MMF appears to underperform in simulating large-scale circulation. Considering that CMT has been shown to significantly modulate large-scale circulation in CAM3.0 (Richter and Rasch, 2008), we hypothesize that this degradation may result from the fact that the convective parameterization in CAS-ESM accounts for CMT effects, whereas the CRM embedded in CAS-ESM_MMF does not. In the following section, we compare CAS-ESM_MMF simulations with and without CMT representation to investigate its impact on large-scale circulation and precipitation within the MMF framework.

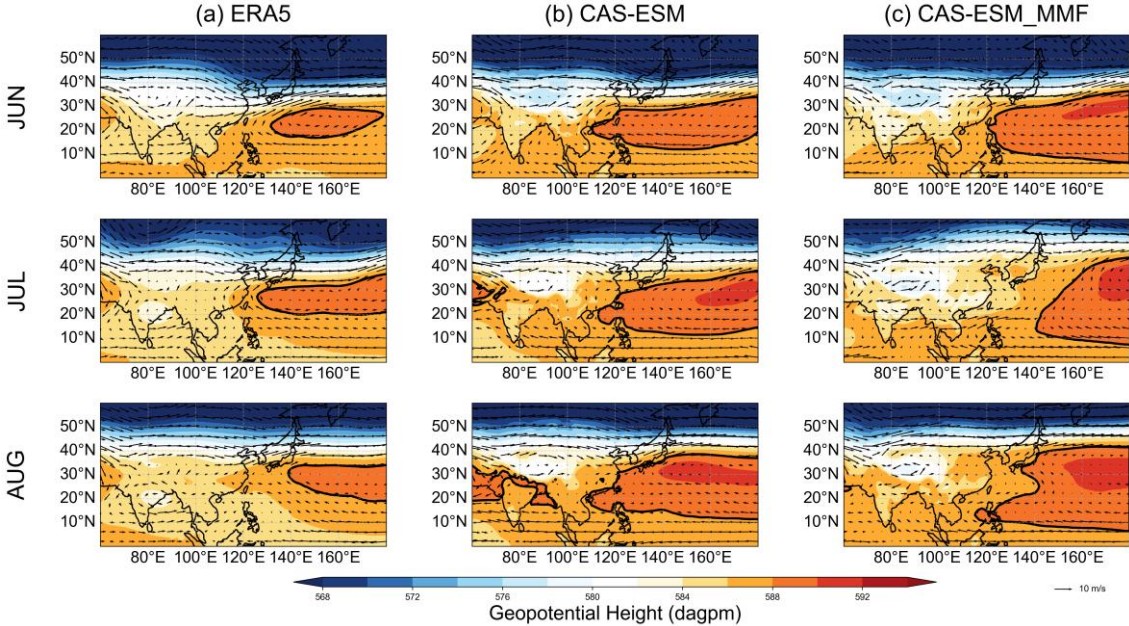

**Figure 8. 6-year (2001-2006) averaged 500 hPa wind vector and geopotential height (shaded) (unit: dagpm) of summer months from (a) ERA5, (b) CAS-ESM, (c) CAS-ESM_MMF. The bold solid line represents 588 dagpm contour line.**



**3.5 Impact of CMT on western Pacific subtropic high and East Asian rainband northward movement**

Lu and Dong (2001) reported a significant negative correlation between the westward extension of the Western Pacific Subtropical High (WPSH) and suppressed warm pool convection. Consistent with this relationship, Luo and Stephens (2006) showed that the Colorado State University (CSU) MMF, which is based on a 2D CRM without momentum feedback, exhibited overly strong convection in the western Pacific. This bias was manifested in anomalously strong precipitation, enhanced surface winds, and increased evaporation, which were tightly coupled through a strengthened "convection–wind–

evaporation" feedback loop. Similar results were found in the Goddard MMF, which used a different CRM and host-GCM (Tao et al., 2009), indicating a persistent issue in MMFs that neglect CMT. Cheedela and Mapes (2019) argued that CMT typically acts to damp vertical wind shear over tropical oceans, thereby weakening the positive feedback that links deep convection, surface wind enhancement, and evaporation. The absence of this damping effect in the MMF likely contributes to the over-amplification of convection in the western Pacific. Supporting this, Khairoutdinov et al. (2005) demonstrated that

incorporating momentum feedback in a 3D-CRM version of SP-CAM reduced the precipitation bias over the western Pacific. Yang et al. (2022) further showed that parameterizing CMT in a 2D CRM can reproduce momentum transport fields similar to those from fully 3D CRMs. Taken together, these studies suggest that neglecting CMT in the CAS-ESM_MMF likely leads to excessive convection over the western Pacific, which in turn biases the simulated WPSH position and intensity.

To test the aforementioned hypothesis, we conduct an additional simulation (CAS-ESM_MMF_MF) that incorporates the

explicit scalar momentum transport (ESMT) scheme to parameterize CMT in the CAS-ESM_MMF. We then compare this simulation to the original CAS-ESM_MMF experiment, which does not include CMT. Considering CAS-ESM_MMF_MF only runs 4 years, the observations and CAS-ESM_MMF data from 2001 to 2003 are used. We first examine changes in the seasonal migration of the East Asian rainband in response to the inclusion of CMT. Figure 9 shows the northward migration characteristics of the rainband, following the same format as Fig. 7. Both the original CAS-ESM_MMF and the CMT-

enabled CAS-ESM_MMF_MF capture the primary seasonal pattern of northward advancement during monsoon onset and southward retreat during monsoon withdrawal. However, in the original CAS-ESM_MMF, the rainband's northward shift begins prematurely in May, a full month earlier than observed. The inclusion of CMT in CAS-ESM_MMF_MF delays the onset of this migration to June, aligning more closely with observations. Moreover, while the original simulation exhibits an abrupt northward jump of the rainband in June, the CMT-enabled simulation displays a smoother, more gradual transition,

which better reflects the observed seasonal evolution. Despite these improvements, the mean latitudinal position of the rainband in CAS-ESM_MMF_MF remains approximately 4° too far north.



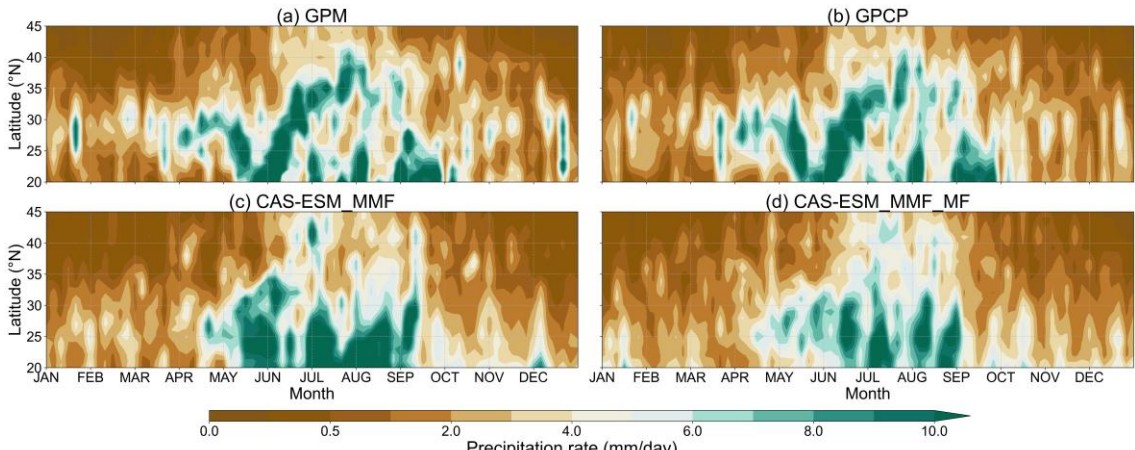

**Figure 9. Similar to Figure 7 but presents results from (a) GPM, (b) GPCP-1dd, (c) CAS-ESM_MMF, (d) CAS-ESM_MMF_MF. The daily climatology is calculated from 2001 to 2003.**

Figure 10 compares the observed and simulated July mean precipitation over East Asia. Given that precipitation in low-latitude regions is primarily driven by deep convection, it serves as a direct proxy for convective intensity. We focus on July because this is when the WPSH bias is most pronounced in the original CAS-ESM_MMF. As shown in Figs. 10b and c, the original simulation strongly overestimates convective precipitation across the Indian subcontinent, Indochina Peninsula, South China, and the western Pacific, resulting in a widespread wet bias. After introducing the CMT parameterization (Figs.

10 d and e), convective intensity in these regions is notably reduced, significantly alleviating the wet bias.



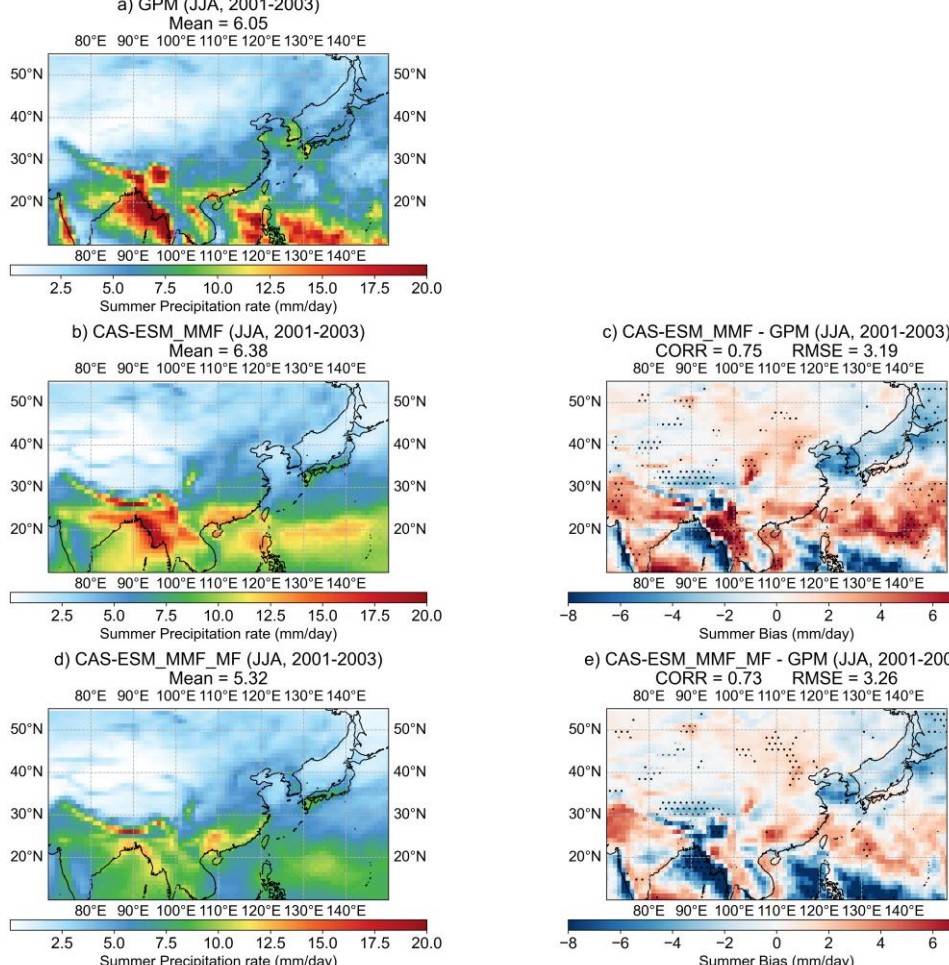

**Figure 10. JJA mean precipitation (mm/day) for 2001-2003 from (a) GPM observation, (b) CAS-ESM_MMF, and (d) CAS-ESM_MMF_MF models. Panels (c) and (e) show the model biases relative to GPM. Area-weighted means are noted in (a, b, d). Spatial correlation (CORR) and RMSE are shown for bias plots. Stippling indicates where the bias is statistically significant (p < 0.05, Student's t-test).**

To further investigate the impact on large-scale circulation, Figure 11 presents the 500 hPa wind and geopotential height fields for June, July, and August. The introduction of CMT leads to a more realistic simulation of the WPSH, improving both its location and intensity. These changes confirm that the inclusion of momentum feedback not only reduces convective biases but also contributes to better representation of key circulation features that govern East Asian summer precipitation.



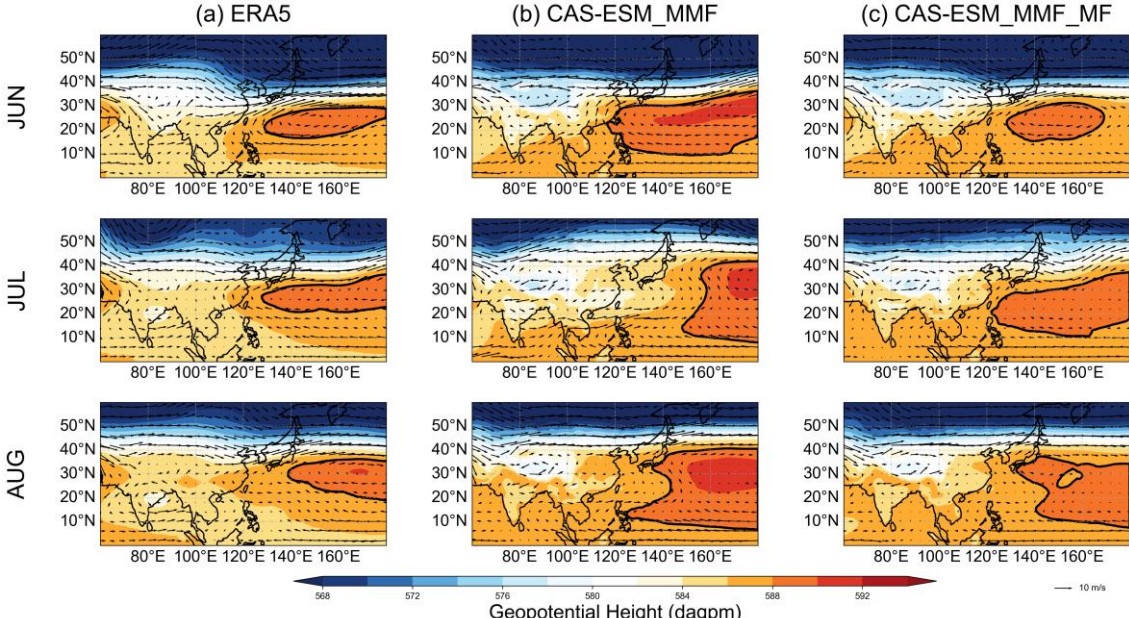

**Figure 11. 3-year (2001-2003) averaged 500 hPa wind vector and geopotential height (shaded) (unit: dagpm) of summer months from (a) ERA5, (b) CAS-ESM_MMF, (c) CAS-ESM_MMF_MF. The bold solid line represents 588 dagpm contour line.**

## 4 Summary and discussion

In this study, we develop a Multiscale Modeling Framework (MMF) version of CAS-ESM (CAS-ESM_MMF), which replaces the traditional convective parameterization scheme in CAS-ESM with a 2D cloud resolving model (SAM) to improve the simulation of East Asian precipitation. The results demonstrate that the CAS-ESM_MMF significantly improves precipitation simulation over East Asia in multiple aspects.

The CAS-ESM_MMF alleviates persistent summer precipitation biases found in the CAS-ESM, namely, the underestimation of rainfall over South China and along the eastern coast of the Bay of Bengal, and the overestimation along the eastern periphery of Tibetan Plateau. Sub-regional analyses show that the seasonal evolution of precipitation simulated by the CAS-ESM_MMF aligns more closely with observations compared to the CAS-ESM. An intensity–frequency analysis of hourly precipitation in key sub-regions reveals that the CAS-ESM tends to produce too much light rain and insufficient heavy rain over South China and the lower Yangtze River basin. The CAS-ESM_MMF substantially reduces this bias, bringing the distribution of precipitation intensity closer to observations. Our results are consistent with those of Zhang and Chen (2016), who reported more realistic frequency–intensity relationship and reduction of precipitation bias in eastern Tibetan Plateau and southern China in super-parameterized CAM5.

CAS-ESM_MMF also improves the representation of intraseasonal variability, particularly the northward migration of the East Asian summer rainband, although it exhibits earlier monsoon rainfall onset and faster progression. Similar findings have



been documented by Jin and Stan (2016). Large-circulation analysis indicates that in the CAS-ESM, an overly strong WPSH

contributes to the dry bias in South China, whereas in the CAS-ESM_MMF, an underestimated WPSH is associated with the

bias in northward movement of rainband. We propose that this underestimation of the WPSH in the CAS-ESM_MMF stems

from the neglect of convective momentum transport (CMT), a common limitation in MMFs employing two-dimensional

CRMs. To evaluate this hypothesis, we implemented a CMT parameterization scheme based on explicit scalar momentum

transport (ESMT). The results show that including CMT suppresses excessive convection over the western Pacific, leading

to a more realistic representation of the WPSH and, consequently, a more accurate simulation of the East Asian summer

rainband.

Overall, the results of this study show that MMF helps improve the simulation of East Asia summer precipitation in many

aspects, such as climatology, annual cycle, PDF, and rainband movement. However, due to neglection of CMT, CAS-

ESM_MMF remains to have moderate biases of precipitation and large-scale circulation. Introduction of CMT

parameterization (ESMT scheme) alleviates these biases and further improves the simulation of rainband movement.

Despite these improvements, some challenges remain. For instance, a notable wet bias persists on the eastern periphery of

the Tibetan Plateau in CAS-ESM_MMF and CAS-ESM_MMF with CMT. This bias is likely due to the coarse grid

resolution of its host-GCM (~100 km), which struggles to adequately resolve the dynamical and thermal forcing associated

with the complex terrain. Future work should consider incorporating a representation of subgrid-scale topography within the

embedded CRM, aiming for a more accurate depiction of orographic precipitation.

In addition, this study has shown that introducing a CMT parameterization in CAS-ESM_MMF can effectively improve the

representation of rainband migration by modulating the interaction between convection and the large-scale circulation. The

physical mechanism underlying this improvement remains to be fully understood, but may relate to the theoretical

framework of Gill (1980), which describes the atmospheric response to tropical heat sources. Further investigation is needed

to validate this hypothesis and elucidate the associated dynamics.

*Data availability*. The GPM Final Run V06B L3 data set can be downloaded from https://doi.org/10.5067/GPM/IMERG/3B-
HH/06 (Precipitation Processing System (PPS) At NASA GSFC, 2022) and the GPCP V2.3 data set at
https://doi.org/10.7289/V56971M6 (Adler et al., 2017) and GPCP 1-degree daily (1dd v1.3) product is accessible at
https://doi.org/10.7289/V5RX998Z (Adler et al., 2017). ERA5 reanalysis data set is accessible via
https://doi.org/10.24381/CDS.143582CF (Copernicus Climate Change Service, 2023).

*Code availability*. The model source code, simulation results, and plotting scripts of CAS-ESM, CAS-ESM_MMF, and
CAS-ESM_MMF_MF can be obtained at https://doi.org/10.5281/zenodo.16910953 (Lin and Liao, 2025).

*Author contribution*. GL designed and led the project. GL and WK developed the model source code with substantial
contributions from ZL and HZ. WK performed model simulations. WL performed analysed data, and prepared the figures
with contributions from XG. GL and WL wrote the paper. ZX, QY, CW, and MZ contributed to the interpretation of results
and improvement of the paper.

*Competing interests*. The authors declare that they have no conflict of interest.



*Acknowledgments.* This study is supported by the National Key Research and Development Program for International Scientific and Technological Innovation Cooperation (grant 2022YFE0195900) and the National Natural Science Foundation
of China (42275086, 42275173). We thank for the technical support of the National Large Scientific and Technological Infrastructure "Earth System Numerical Simulation Facility" (https://cstr.cn/31134.02.EL). During the preparation of first draft, the authors used ChatGPT in order to improve language. After using this tool, the authors reviewed and edited the content as needed and take full responsibility for the content of publication.

*Financial support.* This study is supported by the National Key Research and Development Program for International
Scientific and Technological Innovation Cooperation (grant 2022YFE0195900) and the National Natural Science Foundation of China (42275086, 42275173).

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
