# Peer review of "Development of CAS-ESM\_MMF: Improving East Asian Summer Precipitation Simulation with a Multiscale Modeling Framework"

_EGUsphere, 2025_

## Author Comment (AC1)

We thank both reviewers for their thorough review and insightful comments, which have improved the quality of our manuscript. We have carefully considered all of their suggestions and made every effort to address them. In this point-by-point response, we provide supplementary analyses to address the reviewers' comments, including a diagnostic simulation related to CMT mechanisms and an evaluation of the precipitation diurnal cycle.

Accordingly, in the revised manuscript, we have:

- Incorporated discussions to reflect these new insights and acknowledge model limitations.

- Improved the clarity of figure presentations, such as by updating color schemes and adding visual guides.

- Corrected specific errors and refined wording in response to the reviewers' suggestions.

Below, we have included the reviewers' original comments, each followed by our responses in blue. All line and figure references refer to the revised version of the manuscript.

**Reviewer #1**

This paper, "Development of CAS-ESM_MMF: Improving East Asian Summer Precipitation Simulation with a Multiscale Modeling Framework," presents the development and evaluation of a Multiscale Modeling Framework (MMF) version of the CAS-ESM model. By embedding a cloud-resolving model into the atmospheric component of CAS-ESM, the study demonstrates substantial improvements in simulating East Asian summer precipitation—particularly in alleviating wet and dry biases and in better capturing the seasonal evolution and northward migration of the East Asian rainband. The authors further enhance the framework by incorporating a convective momentum transport (CMT) parameterization, which leads to more realistic large-scale circulation and rainfall migration characteristics.

Under the current background of reduced maintenance and uncertain continuity of the E3SM-MMF codebase, this work represents a timely and valuable step toward advancing operational MMF-based climate simulations. The paper is well-structured and methodologically sound. I think the paper will be a valuable contribution to the MMF modeling community.

**Specific Comments**

**Clarification of CMT Implementation (Section 3.5):**
Please clarify whether, in the CAS-ESM_MMF_MF experiment, the convective momentum transport on the GCM side (Richter & Rasch 2008) remains active or is fully replaced by the ESMT parameterization. This distinction is essential to interpret whether the CMT effects arise solely from the embedded CRM or from combined GCM- and MMF-level treatments.、

Reply: In CAS-ESM_MMF, the deep convection parameterization in the host GCM is turned off. Consequently, the GCM-side CMT associated with the convection scheme is not active; CMT is represented via the ESMT parameterization that complements the embedded 2-D CRM. We have added a clarifying sentence in Section 3.5 (Line 312-314: "Note that deep convection parameterization in CAS-ESM_MMF is turned off. Consequently, there is no CMT effects from deep convection parameterization in CAS-ESM_MMF…") stating that momentum feedback arises from the CRM (via ESMT) rather than from a concurrent GCM convection scheme.

**Line 118:**
Remove the extra space before the comma in "momentum feedback , namely."
Reply: We have removed this. Thanks.

**Line 170:**
The sentence "Overall, the MMF improves the simulation of convective precipitation on a global scale, though some systematic errors remain." seems overstated, since the CORR and RMSE values in Figure 2e appear less favorable than those in Figure 2c. Please rephrase to acknowledge the slight degradation in global statistics despite regional improvements.

Reply: Thanks for pointing this out. We have reorganized the wording in the manuscript to clearly state this point and to distinguish visual pattern similarity from objective statistical performance. (Line 171-174: "Overall, the MMF shows regional improvements in precipitation simulation and appears visually closer to observations while the global spatial correlation and root-mean-square error reveal slight degradation, indicating that CAS-ESM_MMF retains some systematic biases that worth further model tuning.")

**Line 258:**

Since "eastern China" is newly introduced as a defined analysis region, please mark this region's location on the maps in Figure 6 for clarity.

Reply: We have boxed the eastern China analysis region in every subpanel of revised Figure 6 and mentioned this explicitly in the figure caption.

**Lines 325–330:**

The statement that "After introducing the CMT parameterization … convective intensity … is notably reduced" is supported by Figure 10; however, the new simulation appears drier than observations in several regions. Please discuss this residual dry bias and, if feasible, whether modest tuning could better balance the overall magnitude of precipitation.

Reply: We acknowledge that the CAS-ESM_MMF_MF simulation exhibits residual dry biases relative to observations. This dryness is particularly evident over the Bay of Bengal and the South China Sea, where precipitation is also underestimated in the CAS-ESM_MMF run, though to a lesser degree. The introduction of CMT further weakens rainfall over these regions. A plausible explanation is that the CMT exerts damping effects on low-level winds, leading to weaker convergence and thus less precipitation (Please refer to Figure R4 in our reply to Reviewer #2. We have also pasted the figure below for easy reference). At this stage, we have not applied any parameter retuning in this study in order to isolate the physical impact of CMT itself. Further experiments with modest parameter adjustments may help refine the overall precipitation magnitude, but such tuning is beyond the scope of the current work and will be explored in future research. We have added discussion in the revised manuscript. (Line 334-339: "…CMT reduces the overall precipitation rate in low-latitude regions and a possible reason for this reduction is that CMT generally imposes frictional effects on wind (Cheedela and Mapes, 2019), thereby reducing low-level convergence and precipitation…")

[Figure]

Figure R4. Two-year averaged July wind components (a, c) on 850 hPa and wind tendencies due to CMT (b, d).

**Figures 4d & 5d:**

Revise the title and references to "**the middle and lower reaches of the Yangtze River Basin**" for consistency with Figure 1 and the regional definitions used throughout.

Reply: We have revised the title for consistency.

**Figure 6d:**

The northward propagation of the rainband is not as visually clear as in panels (a–c). Consider adding a dashed line or similar visual guide to indicate its seasonal progression.

Reply: We now indicated the rainband path with arrows in Fig. 6d and noted this in the text (Line 256: "…indicated by red arrows in Fig.6d…").

**Figure 9:**

The northward propagation of the rainband in the CAS-ESM_MMF_MF run is less clear than in the observational panels and the original MMF run. Please consider adding a visual guide (e.g., a dashed ridgeline of maximum zonal-mean precipitation or a shaded latitude band for the top decile).

Reply: Thank you for the suggestion. To better visualize the rainband's seasonal progression, we have revised Figure 7 and 9 to include a thick black 5 mm/day contour line as a visual guide. This clearly highlights the rainband's position and movement in all panels. And we also add descriptions about 5 mm/day contour line in the figure caption.

In addition, CAS-ESM_MMF_MF shows local monthly peaks from May to September that are not evident in Figure 7d. Could the authors discuss whether these intra-month oscillations might be related to the inclusion of CMT/ESMT?

Reply: We believe that the reviewer refers to the region around 22° N–28° N in Fig. 9d. We analyzed the time series of area-averaged precipitation over this region using a power-spectrum analysis. The results show that in CAS-ESM_MMF_MF (orange line in Figure R1), the spectral power in the 10–20-day period band is lower than in CAS-ESM_MMF and also lower than in the observations. Therefore, these local monthly peaks are not unique to CAS-ESM_MMF_MF; they also exist in CAS-ESM_MMF but appear less pronounced because of the colorbar scaling in the figure. We have added discussion about this in the revised manuscript (Line 325-328: "Additionally, Fig 9d appears to show intra-month oscillation from May to September. Power-spectrum analysis of the time series of domain averaged precipitation shows that the intra-month oscillation is not unique in CAS-ESM_MMF_MF…")

[Figure]

Figure R1. Power spectrum density of area-mean (22° N-28° N, 110° E-130° E) daily precipitation.

**Reviewer #2**

In the manuscript, the authors developed a Multiscale Modeling Framework (MMF) by implementing a 2-D cloud-resolving model SAM into the Chinese CAS Earth system model (CAS-ESM) and compared its performance over China in the simulation of East Asian summer monsoon with the conventional version of the CAS-ESM. They further included convective momentum transport (CMT) in the MMF as a further improvement of the model's capability. Their results showed that the MMF improved the annual variation of precipitation in several selected regions over China. The precipitation intensity pdf is also better simulated. In the monsoon region (south China and east China) the northward progression of monsoon precipitation is captured in the MMF, especially the one including CMT. Overall, the improvements appear modest, but the capacity building itself is worth documenting. The paper is well written, and the topic fits the scope of GMD. I recommend minor revisions before acceptance.

**Major comments**

One of the main features of summer precipitation over China besides the north-south progression of the monsoon rain is the eastward propagation of the nocturnal precipitation systems in the foothills of the Tibetan Plateau. Is it captured in the CAS-ESM-MMF? A composite diurnal cycle using time-longitude hovmuller plot for region b) should be able to show this.

Reply: We appreciate the reviewer for highlighting this crucial feature of the East Asian summer monsoon. Following the methodology of Zhang and Chen (2016), we examined the diurnal cycle of precipitation along the eastern foothills of the Tibetan Plateau (TP) and plotted the corresponding time-longitude Hovmöller diagram (see Figure R2). As shown in the analysis, neither CAS-ESM nor CAS-ESM_MMF adequately reproduces the observed diurnal evolution characteristics. Specifically, both models struggle to capture the correct timing of the precipitation maximum and fail to simulate the eastward propagation of nocturnal rainfall from the foothills toward the plains. We hypothesize that this deficiency may stem from a combination of unresolved processes, including complex topographic forcing, the interactions with low-level jets, and nocturnal boundary-layer dynamics. While improving this feature is beyond the scope of the current study, we have added a discussion regarding this limitation in the revised manuscript (Line 390-392: "…Furthermore, our analysis of the diurnal cycle indicates that the model fails to capture the eastward propagation of nocturnal precipitation systems from the TP foothills to the plains…").

*Reference*
*Zhang, Y. and Chen, H.: Comparing CAM5 and Superparameterized CAM5 Simulations of Summer Precipitation Characteristics over Continental East Asia: Mean State, Frequency–*

*Intensity Relationship, Diurnal Cycle, and Influencing Factors, Journal of Climate, 29, 1067–1089, https://doi.org/10.1175/JCLI-D-15-0342.1, 2016.*

[Figure]

Figure R2. Diurnal–zonal distributions of the normalized (by daily mean) precipitation amount averaged within 28° N–35° N.

The inclusion of convective momentum transport (CMT) shows some interesting improvements in the monsoon simulation. What are the physical or dynamical mechanisms that led to such improvements? Fig. 11 shows the 500 hPa geopotential height and circulation changes. How are they related to CMT? Can the authors look at the tendencies of winds due to CMV and whether they can explain the circulation differences in the monsoon region?

Reply: We thank the reviewer for this constructive suggestion regarding the physical mechanisms of CMT.

Since our original simulation did not output the explicit wind tendencies due to CMT, we conducted a supplementary two-year simulation which outputs CMT tendencies to address this question.

We analyzed the precipitation, low-level wind fields, and the explicit CMT wind tendencies. The mechanism can be summarized as follows:

1. Bias in the Non-CMT Run: In the original CAS-ESM_MMF, the low-level winds (850 hPa) over the Philippines are excessively strong. This leads to overly strong zonal wind convergence northeast of the Philippines, which in turn sustains excessive convective precipitation in this region (as shown in Figure R3b).

2. Direct Dynamical Effect of CMT: The explicit tendency analysis reveals that CMT exerts a damping effect (frictional drag) on the low-level winds over the Western North Pacific (WNP). This parameterization effectively decelerates the low-level flow, thereby weakening the zonal wind convergence (Figure R4b).

3. Thermodynamic Feedback and Large-Scale Circulation: The reduction in wind convergence directly suppresses the excessive convection and precipitation. According to the framework of Rodwell and Hoskins (2001), the formation of the western and southern boundaries of the subtropical anticyclone is largely a response to the diabatic heating associated with monsoon precipitation (a Gill-type response, Figure R3e). By alleviating the excessive precipitation (and thus the latent heat release) over the WNP, the CMT indirectly corrects the location and intensity of the Western Pacific Subtropical High (WPSH).

In summary, CMT improves the monsoon simulation through a chain of processes: damping low-level winds → reducing moisture convergence → suppressing excessive latent heating → correcting the Gill-type response of the WPSH. We have added a discussion of this physical mechanism in the revised manuscript (Line 380-385: "Further diagnostic analysis of the wind tendencies (figure not shown) suggests that the CMT parameterization exerts a frictional drag on the low-level zonal winds over the Western North Pacific…")

*Reference*
*Rodwell, M. J. and Hoskins, B. J.: Subtropical Anticyclones and Summer Monsoons, J. Climate, 14, 3192–3211, https://doi.org/10.1175/1520-0442(2001)014<3192:SAASM>2.0.CO;2, 2001.*

[Figure]

Figure R3. Two-year averaged July 850 hPa wind field and precipitation rate averaged from 2001 to 2003 from (a) ERA5 reanalysis, (b) CAS-ESM_MMF, and (c) CAS-ESM_MMF_MF. Last two subplots are the differences of 500 hPa wind field and precipitation between (d) CAS-ESM_MMF and ERA5 and (e) CAS-ESM_MMF_MF and CAS-ESM_MMF respectively.

[Figure]

Figure R4. Two-year averaged July wind components (a, c) on 850 hPa and wind tendencies due to CMT (b, d).

**Minor comments**

Fig. 2 and several other figures. The color scheme for the difference plots is opposite to what is typically used in the literature. While it's more of a personal preference, I'd suggest the authors use blue color for positive precipitation biases and red for negative biases for ease of reading.

Reply: We appreciate the reviewer's suggestion regarding the readability of the difference plots. To address this and ensure the most intuitive visualization for precipitation anomalies, we have updated Figures 2, 3, and 10 using a diverging green-brown color scheme. In this scheme, green tones represent positive biases (wetter conditions) and brown tones represent negative biases (drier conditions). We believe this provides a clear and intuitive "wet-dry" contrast that facilitates interpretation.

L159. The SST is a climatological mean over 1995-2005. Can the simulation be called 2001-2006 simulation? I understand that Jan. 1, 2000 atmospheric conditions were used for model initialization, but the initial atmospheric state information should be lost quickly.

Reply: Please refer to our detailed response to the comment below (regarding Fig. 6)

Fig. 6. Is the GPM average also over 2001-2006? As mentioned above, since SST is a climatological mean, comparing model simulations with 2001-2006 observations may not be a consistent comparison.

Reply: We thank the reviewer for pointing out this inconsistency. We acknowledge that comparing a simulation forced by climatological SSTs (1995-2005 mean) with 2001-2006 observations is not strictly consistent. Ideally, the model output should be compared with the

observational climatology covering the same 1995-2005 period. However, the GPM precipitation dataset, which provides the high-quality precipitation data required for this study, is only available from 1998 onward. Since our study focuses on the climatological mean state of the East Asian summer monsoon rather than interannual variability, we assessed whether the 2001–2006 period is representative of the longer climatology. We compared the observed mean precipitation between the available longer period (1998-2005) and the analysis period (2001-2006), as shown in Figure R5. The comparison reveals that though there are slight differences in the tropical oceans (e.g., slightly higher precipitation over the equatorial Pacific and lower over the Maritime Continent during 2001–2006), the differences over East Asia, which is our region of interest, are small and statistically insignificant. Therefore, we believe that using the 2001-2006 mean as a reference provides a reliable benchmark for evaluating the climatological performance of CAS-ESM and CAS-ESM_MMF in this region.

[Figure]

Figure R5. Global annual mean precipitation averaged over 1998-2005 and 2001-2006 respectively and their difference. Stippling indicates where the bias is statistically significant (p < 0.05 Welch's t-test).

L170-171. "Overall, the MMF improves the simulation of convective precipitation on a global scale, though some systematic errors remain." From the global correlation coefficient and the RMSE, MMF appears to have degraded the precipitation simulation instead. So, I wouldn't make this statement.

Reply: Thanks for the reviewer's comment. The first reviewer raised a similar point, and we

have revised the corresponding statement in the manuscript. (Line 171-174 "Overall, the MMF shows regional improvements in precipitation simulation and appears visually closer to observations while the global spatial correlation and root-mean-square error reveal slight degradation, indicating that CAS-ESM_MMF retains some systematic biases that worth further model tuning.")

L94. Delete "that" after "while".

Reply: Revised. Thanks for pointing this out.

L98. Delete "situated".

Reply: We have revised this.

L145. "Analytic"? Do you mean "Analysis"?

Reply: Revised as suggested.

L219. "contribute"

Reply: Revised as suggested.

Please check for grammatic errors in general.

Reply: We appreciate the reviewer's suggestion. We have carefully checked the manuscript for grammatical and typographical errors and have corrected them throughout the text.